# Prevalence of Hepatitis B Virus Markers in Patients with Autoimmune Inflammatory Rheumatic Diseases in Italy

**DOI:** 10.3390/microorganisms8111792

**Published:** 2020-11-16

**Authors:** Marco Canzoni, Massimo Marignani, Maria Laura Sorgi, Paola Begini, Michela Ileen Biondo, Sara Caporuscio, Vincenzo Colonna, Francesca Della Casa, Paola Conigliaro, Cinzia Marrese, Eleonora Celletti, Irene Modesto, Mario Stefano Peragallo, Bruno Laganà, Andrea Picchianti-Diamanti, Roberta Di Rosa, Claudia Ferlito, Simonetta Salemi, Raffaele D’Amelio, Tommaso Stroffolini

**Affiliations:** 1UOC di Immunologia Clinica e Reumatologia, Sapienza Università di Roma, AOU S. Andrea, 00189 Roma, Italy; marialaura.sorgi@uniroma1.it (M.L.S.); biondo_michela@yahoo.it (M.I.B.); sara.caporuscio1@gmail.com (S.C.); vincenzocolonna84@gmail.com (V.C.); francesca.dellacasa4@gmail.com (F.D.C.); bruno.lagana@uniroma1.it (B.L.); andrea.picchiantidiamanti@uniroma1.it (A.P.-D.); roberta.dirosa@uniroma1.it (R.D.R.); clau.ferlito@gmail.com (C.F.); simonettasalemi@gmail.com (S.S.); raffaele.damelio@uniroma1.it (R.D.); 2UOC Malattie Apparato Digerente e Fegato, Sapienza Università di Roma, AOU S. Andrea, 00189 Roma, Italy; paolabegini@virgilio.it; 3UOC di Reumatologia, Dipartimento di “Medicina dei Sistemi”, Università di Tor Vergata, 00133 Roma, Italy; paola.conigliaro@uniroma2.it; 4Ambulatorio di Reumatologia, ASL Roma 1, Presidio Nuovo Regina Margherita, 00153 Roma, Italy; paoloeci@inwind.it; 5Istituto di Clinica Medica, ASL Lanciano-Vasto-Chieti, 66100 Chieti, Italy; cellettieleonora@gmail.com; 6Unità Operativa di Medicina Interna, Università degli Studi di Palermo, AO Ospedali Riuniti Villa Sofia-Cervello, PO Vincenzo Cervello, 90146 Palermo, Italy; irenemodesto01@gmail.com; 7Centro Studi e Ricerche di Sanità e Veterinaria, Esercito Italiano, 00184 Roma, Italy; msperagallo@libero.it; 8Dipartimento di Malattie Infettive e Tropicali, Policlinico Umberto I, 00161 Roma, Italy; tommaso.stroffolini@hotmail.it

**Keywords:** HBV markers, rheumatic patients, immunosuppressive therapy, HBV vaccine

## Abstract

Chronic hepatitis B virus (HBV) infection may be reactivated by immunosuppressive drugs in patients with autoimmune inflammatory rheumatic diseases. This study evaluates HBV serum markers’ prevalence in rheumatic outpatients belonging to Spondyloarthritis, Chronic Arthritis and Connective Tissue Disease diagnostic groups in Italy. The study enrolled 302 subjects, sex ratio (M/F) 0.6, mean age ± standard deviation 57 ± 15 years, 167 (55%) of whom were candidates for immunosuppressive therapy. The Spondyloarthritis group included 146 subjects, Chronic Arthritis 75 and Connective Tissue Disease 83 (two patients had two rheumatic diseases; thus, the sum is 304 instead of 302). Ten subjects (3%) reported previous anti-HBV vaccination and tested positive for anti-HBs alone with a titer still protective (>10 IU/mL). Among the remaining 292 subjects, the prevalence of positivity for HBsAg, isolated anti-HBc, anti-HBc/anti-HBs, and any HBV marker was 2%, 4%, 18%, and 24%, respectively. A total of 26/302 (9%) patients with γ-globulin levels ≤0.7 g/dL were more frequently (*p* = 0.03455) prescribed immunosuppressive therapy, suggesting a more severe rheumatic disease. A not negligible percentage of rheumatic patients in Italy are at potential risk of HBV reactivation related to immunosuppressive therapy. Before starting treatment, subjects should be tested for HBV markers. Those resulting positive should receive treatment or prophylaxis with Nucleos (t) ides analogue (NUCs) at high barrier of resistance, or pre-emptive therapy, according to the pattern of positive markers. HB vaccination is recommended for those who were never exposed to the virus.

## 1. Introduction

Hepatitis B virus (HBV) infection still represents a relevant threat to global public health, considering that approximately 350 million people are estimated to be affected worldwide [1]. Despite the availability, for over three decades, of an effective vaccine included in the infant vaccination schedule of most countries [2], approximately 2 billion people worldwide have come in contact with the virus [3].

HBV is a DNA virus belonging to the *Hepadnaviridae* family [4]. Following exposure to the virus, HBV-DNA is transformed in covalently closed circular (ccc) DNA, which is integrated in the host genome inside the hepatocyte nucleus [5]. The viral genome encodes for different proteins, of which those used for defining the serological infection profile are the core (HBcAg), surface (HBsAg), and soluble, extractable (HBeAg) antigens, and/or the relative antibodies [4,5]. Nine genotypes and several subgenotypes are currently known [6]. HBV may remain inactive and asymptomatic for a long time in the infected hepatocyte, thus inducing the state of the HBV carrier. According to HBV markers, HBV carriers can be divided into different categories: 1) potential occult HBV infections (pOBIs), characterized by HBsAg negativity, positivity for antibodies to the core antigen (anti-HBc), very low (<200 international units (IU)/mL), or absent HBV-DNA levels, and alanine aminotransferase (ALT) persistently within normal range, unless other potential hepatotoxic agents or conditions (i.e., obesity, alcohol, drugs) may increase this latter test; 2) overt carriers, characterized by HBsAg positivity. This latter category is further subdivided into chronic HBV infection (the former inactive carrier), based on normal or minimally increased ALT value with HBV-DNA persistently below 2000 IU/mL, and chronic HBV hepatitis (the former active carrier), based on the persistence of ALT elevation for at least 6 months, with HBV-DNA higher than 2000 IU/mL in the absence of clinical, biochemical, and ultrasound evidence of liver cirrhosis [6].

Treatment with some immunosuppressive drugs may lead to HBV reactivation [7]. In overt HBsAg carriers, definition of reactivation is based on a ≥ 1 log_10_HBV-DNA increase as compared to the value before immunosuppression, or the de novo HBV-DNA detection in a previously negative subject [7]. pOBI may also experience reactivation in a small percentage of patients undergoing intense immunosuppressive therapy, with serum HBsAg re-expression (seroreversion) representing a relevant virologic and clinical event, associated with re-appearance of active viral replication [8]. In both cases, HBV reactivation might be followed by acute hepatitis, characterized by serum aminotransferase increase, which may remain subclinical or evolve into acute liver failure and death [7]. Risk of reactivation correlates with pretreatment HBV-DNA levels, but it also depends on the immunosuppressive therapeutic agents used and on the duration of treatment [8,9,10]. Usually, immune-mediated liver inflammation and consequent hepatitis develop after immunosuppressive treatment interruption, due to normal immune function restoration [7,10].

Despite the acknowledged risk of viral reactivation, epidemiological data on the prevalence of HBV markers among patients with autoimmune inflammatory rheumatic diseases (AIIRDs) are scarce [11] and completely lacking in Italy.

The aim of this study was to evaluate the prevalence of HBV markers in patients with AIIRD in Italy.

## 2. Materials and Methods

Patients affected by different rheumatic diseases (Spondyloarthritis group, including Psoriatic Arthritis, Ankylosing Spondylitis, Reactive Arthritis, Entero-arthritis, and Undifferentiated Spondyloarthritis; Chronic Arthritis group, including Rheumatoid Arthritis and Chronic Undifferentiated Arthritis; Connective Tissue Disease group, including Systemic Lupus Erythematosus, Sjögren Syndrome, Systemic Sclerosis, Vasculitis, Polymyositis/Dermatomyositis, Undifferentiated Connective Tissue Disease, and Polymyalgia Rheumatica), candidates for either non-immunosuppressive or conventional synthetic and/or biological immunosuppressive therapy, were consecutively enrolled from January 2012 to June 2013 at Sapienza Università di Roma, AOU S. Andrea, UOC di Immunologia Clinica e Reumatologia, and at Università di Tor Vergata, Policlinico Tor Vergata, UOC di Reumatologia, both in Rome, Italy. Exclusion criteria were: having received corticosteroids at a dose of ≥7.5 mg/day of prednisone or equivalent and/or conventional synthetic Disease-Modifying Anti-Rheumatic Drugs (csDMARDs) in the 6 months preceding the enrolment, and/or having been treated with biologic DMARDs (bDMARDs) at any time, and/or being seropositive for human immunodeficiency virus (HIV). Disease duration in years was calculated from the date of reported start of symptoms and not from the date of official diagnosis, frequently occurring years after the start of symptoms.

Clinical and anamnestic data were collected, including demographic characteristics, diagnosis according to internationally recognized classification criteria, history of previous HBV infection and HBV vaccination, and past and current therapy. Subjects were tested for HBV markers (HBsAg, HBeAg, anti-HBc, anti-HBe, and anti-HBs). Those resulting HBsAg-positive and/or anti-HBc-positive, were subsequently tested for HBV-DNA.

Therapy was prescribed according to the level of immunological disease severity and categorized as follows: non-immunosuppressive therapy, corresponding to non-steroidal anti-inflammatory drugs, low-dose steroids (≤7.5 mg/day of prednisone or equivalent), sulfasalazine or hydroxychloroquine; conventional synthetic immunosuppressive therapy corresponding to methotrexate, leflunomide, cyclosporine A, and azathioprine (high-dose steroids (≥7.5 mg/day of prednisone or equivalent) were also included in this group); biologic immunosuppressive therapy, corresponding to treatment with TNFα inhibitors, excepting one patient who was prescribed abatacept.

All subjects gave their informed consent for inclusion before they participated in the study. The study was conducted in accordance with the Declaration of Helsinki, and the protocol was approved by the Ethics Committee of the S. Andrea University Hospital and the Policlinico Tor Vergata (advice 84/2010).

### 2.1. Laboratory Assay

HBV markers were checked by CLIA (Chemiluminescence ImmunoAssay (Architect)). HBV-DNA was detected by Polymerase chain reaction (PCR) amplification, with a detection limit of 10 IU/mL.

Serum protein electrophoresis, C-reactive protein (CRP) and erythrocyte sedimentation rate (ESR) were carried out with the standard laboratory method.

### 2.2. Statistical Analysis

Differences in proportions were evaluated by two-tail, Yates corrected, χ^2^ or Mid-*p* exact test, when indicated. Continuous variables were compared using the Student’s t test, after natural logarithm transformation, in case of non-normal distribution (this was the case for disease duration and CRP). A *p* value < 0.05 was considered to be significant.

## 3. Results

Three hundred and two patients enrolled. M to F sex ratio was 0.6, the mean age ± standard deviation was 57 ± 15 years, and 167 subjects (55%) were prescribed immunosuppressive therapy (119 conventional synthetic and 48 biologic). Two hundred and seventy-six (91%) were Italians and 26 (9%) were non-Italians; 235 (77%) Italian patients came from North-Centre of Italy and 41 (14%) from South Sicily. The Spondyloarthritis group was formed by 146 (48%) patients, including 91 (30%) Psoriatic Arthritis, 22 (7%) Ankylosing Spondylitis, 6 (2%) Reactive Arthritis, 4 (1%) Entero-arthritis and 23 (8%) Undifferentiated Spondyloarthritis; the Chronic Arthritis group was formed by 75 (25%) patients, including 64 Rheumatoid Arthritis plus 1 Adult Still’s disease (22%) and 10 (3%) Undifferentiated Chronic Arthritis; the Connective Tissue Disease group was formed by 83 (27%) patients, including 10 (3%) Systemic Lupus Erythematosus, 15 (5%) Sjögren Syndrome, 19 (6%) Systemic Sclerosis, 3 (1%) Vasculitis, 3 (1%) Polymyositis/Dermatomyositis, 14 (5%) Undifferentiated Connective Tissue Diseases, and 19 (6%) Polymyalgia Rheumatica. The sum of the patients of the diagnostic groups is 304 instead of 302, because two patients have been counted twice, the first being affected by Psoriatic Arthritis and Vasculitis and the other by Rheumatoid Arthritis and Polymyalgia Rheumatica. No significant difference was observed for disease duration between the different diagnostic groups. The Chronic Arthritis group showed statistically significantly higher CRP and ESR mean values as compared to Connective Tissue Disease (*p* = 0.005) and to Spondyloarthritis (*p* = 0.003), respectively (Table 1).

Ten subjects (3%) reported previous HB vaccination and resulted anti-HB-positive alone, with a still protective value (>10 mIU/mL), 7 of whom belonging to the Connective Tissue Disease and 3 to the Spondyloarthritis group (*p* = 0.03401, Mid-*p* exact test). Among the remaining 292 subjects, the prevalence of positivity for HBsAg, isolated anti-HBc, anti-HBc/anti-HBs, and any HBV marker was 2%, 4%, 18%, and 24%, respectively. No HBsAg-positive subject was also HBeAg-positive (Table 2).

Out of the 70 patients positive for any HBV marker, 21 (30%) were aware of their condition. Among the six overt carriers, the HBV-DNA level was >2000 IU/mL in one case, <2000 IU/mL in another case, and <200 IU/mL or undetectable in the remaining four cases. Among the isolated anti-HBc and anti-HBc/anti-HB-positive subjects, HBV-DNA was <200 IU/mL in four (2 isolated anti-HBc and 2 anti-HBc/anti-HBs positive subjects) and undetectable in all the other subjects.

The prevalence of any HBV marker peaked in the group of Chronic Arthritis as compared to the other diagnostic groups, but the difference was significant (*p* = 0.01146) only vs. Connective Tissue Disease group. Moreover, the prevalence of any HBV markers was higher in subjects older than 60 years as compared to the 60-year-old or younger ones (29% vs. 19%; *p* < 0.05). No statistical difference was observed by sex and by potential suitability for immunosuppressive therapy (Table 3).

In total, 26 out of the 302 (9%) patients had mild hypogammaglobulinemia (levels of gamma globulins ≤0.7 g/dL). Hypogammaglobulinemic patients were more frequently (*p* = 0.03455) prescribed immunosuppressive therapy (Table 4).

These patients were not associated with specific demographic, diagnostic, and/or clinical variables (excepting one patient with cryoglobulinemia and pancytopenia, who had the lowest gamma globulin level (0.5 g/dL)), nor to history of immunosuppressive therapy in the previous 6 months (which represented an exclusion criterion) or HBV marker positivity.

## 4. Discussion

These findings evidence that a great proportion (24%) of subjects affected by AIIRD present serological markers of previous exposure to HBV infection, a figure higher than the prevalence rate of the 8.6% observed in over 30,000 first-time Italian blood donors evaluated in the same period of the current study [12]. This difference is not related to the different geographical locations between the two populations. In fact, the patients residing in the North-Centre of Italy were 82% in the study on the first-time blood donors and 77% in the current study; the patients residing in Southern Italy made up 17% and 14% in the two studies, respectively. The lack of a mean age of the first-time blood donors being reported does not allow comparison with the mean age of the current study. Blood donors are a special population exposed to selective procedures, so that they cannot be considered representative of the general population. However, the nearly three-fold higher HBV marker (any marker) prevalence in patients with AIIRD as compared to blood donors (24% vs. 8.6%) suggests an increased risk of exposure to HBV infection in AIIRD subjects, probably reflecting frequent previous iatrogenic contacts. The present finding is in contrast with data of a meta-analysis showing absence of significant different prevalence of HBV markers between AIIRD patients and general population; unfortunately, the only available studies, which were all included in this meta-analysis, were heterogeneous and with a level of evidence 3b [13].

A potential HBV-mediated induction of AIIRD on the basis of molecular mimicry or bystander activation, as suggested even for HB vaccination [14], cannot be excluded.

We acknowledge that data on HBV prevalence might appear outdated as the study was carried out in the years 2012–2013. In Italy, the prevalence of HBV markers in first-time blood donors was 275.9 × 10^5^ in 2009, declining to 143.6 × 10^5^ in 2018 [15]. However, it should be taken into account that it is still too early to observe the effect of HB vaccination (introduced in 1991) in subjects with AIIRD, who have a mean age of 57 years. Indeed, we still need 2 to 3 decades to observe a potential effect of the preventive measure on HBV markers’ prevalence in this population.

The observed prevalence estimates (as in all epidemiological studies) may lack global representativeness as they were collected in a single country—i.e., Italy. In any case they provide a representative picture of Italy, as they were collected in different areas of the country.

Nonetheless, independently of being a special risk population, these patients are at potential risk of reactivation following immunosuppressive therapy, even though there is a different extent of risk, depending on the pattern profile of the positive markers. The reported incidence rate of HBV reactivation in subjects with resolved HBV infection (i.e., anti-HBc positivity with or without anti-HBs positivity) was 1.7% (12/712) in those treated with bDMARDs and 3.2% (10/327) in those treated with csDMARDs [16]. Conversely, the incidence rate of HBV reactivation was 39% (35/89) among HBsAg-positive subjects treated with anti-TNFα therapy [17]. This latter finding confirms the crucial role of TNFα in the host response to HBV infection [18,19].

In this study, the AIIRD patients belonged to three different diagnostic groups; those with Chronic Arthritis had the highest levels of inflammation markers, such as CRP and ESR. We also checked the prevalence of HBV markers in the three diagnostic groups to evidence differences, if any. Actually, HBV infection markers were more likely observed in patients with Chronic Arthritis, who were also those presenting the highest levels of inflammation markers, a condition recently found to be predictive of low response to HB vaccination [20]; this suggests that this group of patients may also clear HBV less efficiently. Finally, Chronic Arthritis and Spondyloarthritis patients, who had higher prevalence of HBV markers compared to Connective Tissue Disease patients, were more frequently planned for biologic therapy, which increases the risk of HBV reactivation.

Management of AIIRD patients who are candidates for immunosuppressive therapy represents a crucial challenge requiring a strict collaboration between specialists in rheumatology and hepatology. The European Association Study for Liver Diseases (EASL) has recently provided guidelines on this topic [6]. Main aspects are plotted in Figure 1 and herein summarized.

All candidates for immunosuppressive therapy should be tested for HBV markers prior to immunosuppression [21]. This is a crucial point, considering that most subjects are not aware they are HBV-infected and many physicians have a low sensitivity to the need of preventing HBV reactivation in patients undergoing immunosuppressive therapy [22]. In the current survey, only 16% (11/70) of subjects with serological evidence of previous exposure to HBV reported being aware of their status, a rate slightly higher than that estimated for the European Union inhabitants [23], but markedly lower than that of USA citizens [24]. HBsAg-positive subjects should receive either a treatment with Nucleos (t) ides analogue (NUCs) in case of chronic HBV hepatitis, or prophylaxis with NUCs in case of chronic HBV infection. Those with isolated anti-HBc positivity should receive prophylaxis with NUCs, if HBV-DNA-positive, or pre-emptive therapy (i.e., monitoring HBsAg every 1–3 months during and after immunosuppression to check potential HBsAg seroreversion, which requires starting anti-viral treatment), if HBV-DNA-negative. Finally, those with serological evidence of resolved infection (anti-HBc-positive/anti-HBs-positive) should receive pre-emptive therapy. Advised NUCs are those at high genetic barrier of resistance (i.e., entecavir or tenofovir) due to the long duration of treatment or prophylaxis [25]; treatment of chronic HBV hepatitis with NUCs should continue until virological response (i.e., HBsAg loss with anti-HBs seroconversion), whereas anti-viral prophylaxis and pre-emptive therapy should continue for at least 12 months (18 months for rituximab) after discontinuation of immunosuppressive treatment [6,26]. The patients herein described have received antiviral therapy or periodical monitoring, based on the serological status and according to the 2012 EASL report [27], and cases of reactivation during immunosuppressive therapy have not yet been observed.

Vaccination for HBV seronegative subjects is recommended [6]. However, this unlimited recommendation is not universally shared, reflecting a different sensitivity towards the issue of HB vaccination in patients undergoing immunosuppressive therapy between hepatologists and rheumatologists. The experts of the EASL recommended vaccination of the seronegative patients undergoing immunosuppressive therapy in the 2012 report [27] and reiterated the statement in the 2017 report [6]. The experts of the European League Against Rheumatism (EULAR) recommend HB vaccination limited to at-risk individuals undergoing immunosuppressive therapy [28], probably for lack of a reliable demonstration of a higher prevalence of HBV markers found in patients with rheumatic diseases as compared to the general population [13]. The lack of sensitivity to this issue is also witnessed by the only 10 (3%) vaccinated subjects in the current study.

Universal vaccination against HBV infection was introduced in Italy in the summer of 1991 for all infants and all 12-year-old adolescents (the latter limited to the first 12 years of the campaign). As a result, in the current year (2020), virtually all Italians aged 0–40 years had been vaccinated against HBV. It means that subjects older than this age, similar to those with AIIRD, are at risk of acquiring the virus, as not covered by the vaccine. The potential aggressive clinical course observed in HBV seropositive subjects in case of viral reactivation [7,29] strongly suggests vaccination [6,30,31], for those still HBV seronegative. Higher vaccine doses [32] are advised in case of lacking/defective immune response to the vaccine. Clinicians should do their best to immunize patients before administration of immunosuppressive therapy, especially if biologic [33]

HB vaccination of subjects with isolated anti-HBc positivity is suggested based on limited data [34]—this rationale appears weak considering that the aim of vaccination is preventing infection, which has apparently already been established. Nevertheless, HBV vaccination in anti-HBc-positive subjects has been tried since the eighties of the last century, when the vaccine became available, generally for diagnostic purposes, in order to rightly interpret the meaning of isolated anti-HBc positivity [35,36,37,38,39,40,41,42,43,44,45]. In fact, when the observed immune response to vaccine was slow or primary, a relatively frequent condition in older studies [37,38,39,43], a false anti-HBc positivity was hypothesized, on the basis of the consideration that a true anti-HBc positivity witnesses a previous encounter with HBV able to prime the immune system, thus a rapid or anamnestic response would be expected; according to this interpretation, the anti-HBc-positive subjects with slow responses are interpreted as false positives and assimilated to seronegative subjects, thus vaccine cycle needs to be completed. Conversely, if the observed immune response was rapid or anamnestic, the anti-HBc positivity was attributed to a resolved infection, with early loss of anti-HBs antibodies; according to this interpretation, these patients only need one booster. Finally, if the immune response was weak or lacking, anti-HBc positivity was attributed to an occult HBV infection in an individual unable to mount an effective immune response [46]. However, such a schematic interpretation may only in part provide the understanding of this complex topic—it does not face the issue of possible vaccine stimulation of an ineffective immune response, unable to clear the chronic infection. In fact, it is well known the role that HBV migration and localization have for inducing a weak immune response, able to produce anti-HBc but unable to clear the virus and maintain memory cells for longtime, thus allowing the virus be tolerated by the immune system or, alternatively, be located in the hepatocyte, invisible to the immune system [47]. It may be hypothesized that vaccination, by providing an adjuvanted antigen in the best immunizing conditions, may strengthen an exhausted immune response and prevent viral reactivation, at least in some anti-HBc-positive subjects, as it has been observed following HBV vaccination of exposed family contacts of chronic liver disease patients [48].

Mild hypogammaglobulinemia (gamma globulins ≤0.7 g/dL) was found in 26/302 (9%) patients. These subjects had no diagnostic, clinical, and/or demographic specific characteristics, nor significant association with previous immunosuppressive treatment, nor a higher percentage of HBV marker positivity. However, they were more frequently prescribed immunosuppressive therapies suggesting that mild hypogammaglobulinemic status could be associated with a more severe rheumatic disease. Although the cause of mild hypogammaglobulinemia is unknown, this serological status might render subjects more susceptible to infections with consequent reactivation of rheumatic disease. In case this serendipitous observation would be confirmed in larger representative populations, the gamma globulin level could be proposed as an additional diagnostic marker to better define and frame the AIIRD.

## 5. Conclusions

In conclusion, the current epidemiological study underlines the relatively high percentage of patients with AIIRD at potential graded risk of HBV reactivation because of immunosuppressive therapy, emphasizing the need of pre-treatment testing for HBV markers, a measure that rheumatologists have not always been aware of [49]. Subjects testing positive for HBV markers should be urgently referred to a specialist for a further assessment and diagnosis of the phase of HBV infection. Strict collaboration between specialists is of paramount importance for the best management of these patients. Vaccination of the seronegative patients is highly recommended, possibly before starting immunosuppressive therapy and, if necessary, through individualized reinforced vaccine schedules.

## Figures and Tables

**Figure 1 microorganisms-08-01792-f001:**
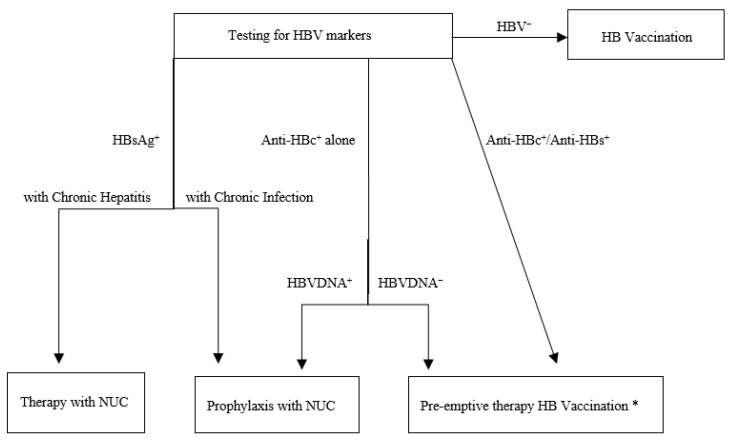
Management of patients with autoimmune inflammatory rheumatic diseases candidate to immunosuppressive therapy (Adapted from reference 6). * Measure suggested on the basis of limited data, in Anti-HBc^+^/HBVDNA^−^ patients.

**Table 1 microorganisms-08-01792-t001:** Demographic and clinical characteristics of 302 enrolled patients.

Characteristic	N	%	DD ^¥^	CRP ^©^	ESR ^∞^
**Sex**					
Male Female Sex ratio M/F	113 189 0.6	63 37			
**Age-groups (years)**					
<40 40–60 >60 Mean age (years)	39 131 132 57	13 43 44			
**Origin**					
Italians	276	91			
North Centre South/Sicily	16 219 41	5 72 14			
Non-Italians	26	9			
**Spondyloarthritis**	146 *	48	4.17 ± 7.47	4.07 ± 9.43	25.74 ± 20.4
**Chronic Arthritis**	75 ^‡^	25	3.36 ± 8.21	5.96 ± 11.33 ^†^	33.88 ± 22.19 ^↕^
**Connective Tissue Disease**	83	27	3.44 ± 9.93	3.82 ± 7.88	34.41 ± 26
**Candidates to immunosuppressive therapy**	167	55			

* One patient had Psoriatic Arthritis and Vasculitis; thus, he was counted twice, in the Spondyloarthritis and Connective Tissue Disease groups; ^‡^ one patient had Rheumatoid Arthritis and Polymyalgia Rheumatica, thus he was counted twice, in Chronic Arthritis and Connective Tissue Disease groups. The sum of the patients results, therefore, was 304 instead of 302; ^¥^ DD = Disease Duration (years, mean ± standard deviation); ^©^ CRP = C-reactive protein (mean ± standard deviation, expressed in mg/dL); ^∞^ ESR = Erythrocyte Sedimentation Rate after first hour (mean ± standard deviation); ^†^
*p* = 0.005 Chronic Arthritis vs. Connective Tissue Disease; ^↕^
*p* = 0.003 Chronic Arthritis vs. Spondyloarthritis.

**Table 2 microorganisms-08-01792-t002:** Prevalence of HBV markers in 292 * patients with autoimmune inflammatory rheumatic diseases.

**HBV Markers**	**N**	%
**HBsAg+ ****	6	2
**Anti-HBc+ alone**	12	4
**Anti-HBc+/Anti-HBs+**	52	18
**Any HBV infection marker positivity**	70	24

* Ten patients excluded because reporting previous HBV vaccination and resulting anti-HBsAg^+^ alone. ** No subject was also HBeAg^+^.

**Table 3 microorganisms-08-01792-t003:** Prevalence of any Hepatitis B virus (HBV) marker according to demographic and diagnostic characteristics.

Characteristic	N Positive/N Tested	%	*p*
**Sex**			
● Male	27/111	24	NS
● Female	43/181	24	
**Age-groups**			
● ≤60	31/159	19	<0.05
● >60	39/133	29	
**Spondyloarthritis**	34/143	24	0.01146 *
**Chronic Arthritis**	25/75	33	
**Connective Tissue Disease**	11/76	14	
**Candidates to immunosuppressive therapy**			
● Yes	35/160	22	NS
● No	35/132	27	

***** Chronic Arthritis vs. Connective Tissue Disease; NS = not significant.

**Table 4 microorganisms-08-01792-t004:** Hypogammaglobulinemic patients and prescribed therapy.

Patients	Non-Immunosuppressive Therapy	Immunosuppressive Therapy	Total
**All** γ-globulins >0.7 g/dL N (%)	129 (47)	147 (53)	276 (91)
**All** γ-globulins ≤0.7 g/dL N (%)	6 (23)	20 (77) *	26 (9)
**SpA** γ-globulins > 0.7 g/dL N (%)	52 (40)	77 (60)	129 (88)
**SpA** γ-globulins ≤ 0.7 g/dL N (%)	5 (29)	12 (71) ^	17 (12)
**ChA** γ-globulins > 0.7 g/dL N (%)	27 (38)	44 (62)	71 (95)
**ChA** γ-globulins ≤ 0.7 g/dL N (%)	1 (25)	3 (75) ^	4 (5)
**CTD** γ-globulins > 0.7 g/dL N (%)	48 (61.5)	30 (38.5)	78 (94)
**CTD** γ-globulins ≤ 0.7 g/dL N (%)	3 (60)	2 (40) ^	5 (6)

* *p* = 0.03455 vs. patients with γ-globulins >0.7 g/dL, by two tails, Yates corrected, χ^2^; ^ Not significant; SpA = Spondyloarthritis; ChA = Chronic Arthritis; CTD = Connective Tissue Disease.

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
