# Peer review of "Prevalence of Hepatitis B Virus Markers in Patients with Autoimmune Inflammatory Rheumatic Diseases in Italy"

_microorganisms, 2020, doi:10.3390/microorganisms8111792_

Round 1
Reviewer 1 Report
This study reports the results of a prospective study on the prevalence of serum markers of HBV infection in an Italian cohort of patients with diverse autoimmune inflammatory rheumatic diseases and candidate to immunosuppressive therapy. In the context of this study, the authors provide a valuable review of prophylaxis/therapy strategies for handling possible HBV reactivation.
The central claim in this manuscript is that “great proportion (24%) of subjects affected by AIIRD present serological markers of previous exposure to HBV infection”, but this issue is scarcely discussed throughout the manuscript. First, there is no comparison with the prevalence of HBV serological markers in the age class-matched general population in Italy to support the statement that patients with AIIRD have an unusually high prevalence. Second, the reason for an unusually high prevalence should be discussed (age distribution? geographical distribution? Possible role for HBV in the pathogenesis of some AIIRDs?).
It would also be of interest to report if patients subsequently treated with immunosuppressive drugs received therapy, prophylaxis, preemptive therapy or vaccination for HBV, and what was the virologic outcome after immunosuppression.
Minor comments
- Page 2 line 77. “some” would be probably better than “given”.
- Page 3 line 105. “corticosteroids at the dose ≥7.5 mg/day” should be changed into “≤7.5 mg/day of prednisone or equivalent” as in line 116 of the same page.
- Page 4 line 162. “no subject” rather than “none subject”.
- Page 6 lines 209-211. Would be useful in the context of the review on guidelines to briefly define therapy and prophylaxis (which drugs? how long?).
Author Response
We wish to thank the reviewers for the constructive remarks which have allowed to substantially improve the manuscript and we apologize for 2 inaccuracies which have been noticed when revising the file, the first represented by the percentage of patients aware of their serological HBV marker situation, which has been reduced from 21/70 (30%) to 11/70 (16%), as a consequence of a more accurate check (line 255), and the second by the changes of 2 numbers in Table 2, which refer to different categories of HBV marker positivity, which do not modify the total amount of HBV positivity. (The new version of the manuscript is attached).
Point 1: The central claim in this manuscript is that “great proportion (24%) of subjects affected by AIIRD present serological markers of previous exposure to HBV infection”, but this issue is scarcely discussed throughout the manuscript. First, there is no comparison with the prevalence of HBV serological markers in the age class-matched general population in Italy to support the statement that patients with AIIRD have an unusually high prevalence. Second, the reason for an unusually high prevalence should be discussed (age distribution? geographical distribution? Possible role for HBV in the pathogenesis of some AIIRDs?).
Response 1: We wish to thank the reviewer for his interesting remark, which allows us to clarify a qualifying point of our study. The reply has been provided in the Discussion, lines 211-229
Point 2: It would also be of interest to report if patients subsequently treated with immunosuppressive drugs received therapy, prophylaxis, preemptive therapy or vaccination for HBV, and what was the virologic outcome after immunosuppression.
Response 2: The reply has been provided for therapy/prophylaxis in the Discussion, lines 270-272; the patients have been administered anti-viral therapy (only 1 HBsAg-positive patient with HBV-DNA 260,000 IU/ml) or periodical HBV markers monitoring, in order to promptly identify any possible sign of reactivation, and in case then proceeding with pre-emptive therapy, according to the EASL 2012 guidelines. Vaccinations have not been administered to patients for a series of bureaucratic difficulties, which have made impossible to proceed to the vaccination even for a small quote of patients. This situation is probably expression of the diffuse anti-vaccination attitude of the general population, especially in developed countries, which has been documented in different occasions.
Point 3: Page 2 line 77. “some” would be probably better than “given”.
Response 3: “Given” has been replaced by “some”.
Point 4: Page 3 line 105. “corticosteroids at the dose ≥7.5 mg/day” should be changed into “≤7.5 mg/day of prednisone or equivalent” as in line 116 of the same page.
Response 4: “corticosteroids at the dose ≥7.5 mg/day” has been replaced by “≥7.5 mg/day of prednisone or equivalent”
Point 5: Page 4 line 162 (now page 5 line 175). “no subject” rather than “none subject”.
Response 5: “None subject” has been replaced by “No subject”
Point 6: Would be useful in the context of the review on guidelines to briefly define therapy and prophylaxis (which drugs? how long?).
Response 6: The drugs were already indicated (entecavir and tenofovir) at line 266, whereas the duration of therapy/prophylaxis has been added at lines 266-270.

Reviewer 2 Report
Canzoni et al present the results of a cross-sectional study evaluating the rates of HBV markers positivity among 302 patients with autoimmune systemic diseases in Italy. The overall prevalence was 24%.
Although well written and presented, It must be pointed out that the study has been conducted in 2012-2013 and the results may be quite outdated. As the Authors correctly state, universal HBV vaccination is nowadays very common in Italy and it has to be taken into account when presenting such dated results. Moreover, the study has been conducted in a small area of Italy and the prevalence estimates may not be very informative for practitioners in other countries.
To improve the importance and informativeness of the study, I would suggest the Authors to update the results, if possibile and available, with patients' follow-up data (eg if they were prescribed immunosuppressive or biological therapy and the outcomes in those with positive HBV markers). Additionally, they could add demographic (eg ethnicity) and clinical data (eg disease activity or disease duration) for completeness in Table 1.
Finally, I do not understand the rationale of comparing proportions between the groups with statistical tests without an a priori hypothesis.
Author Response
We wish to thank the reviewers for the constructive remarks which have allowed to substantially improve the manuscript and we apologize for 2 inaccuracies which have been noticed when revising the file, the first represented by the percentage of patients aware of their serological HBV marker situation, which has been reduced from 21/70 (30%) to 11/70 (16%), as a consequence of a more accurate check (line 255), and the second by the changes of 2 numbers in Table 2, which refer to different categories of HBV marker positivity, which do not modify the total amount of HBV positivity. (The new version of the manuscript is attached).
Authors' Reply to the Review Report (Reviewer 2)
Point 1: Although well written and presented, it must be pointed out that the study has been conducted in 2012-2013 and the results may be quite outdated. As the Authors correctly state, universal HBV vaccination is nowadays very common in Italy and it has to be taken into account when presenting such dated results. Moreover, the study has been conducted in a small area of Italy and the prevalence estimates may not be very informative for practitioners in other countries.
Response 1: We thank the reviewer for his kind appreciation. Actually, the study may appear quite outdated having been realized in 2012-2013 and in Italy the prevalence of HBV markers in first time donors was
275.9×105 in 2009, declining to 143.6×105 in 2018 (Blood Transfus 2019; 17: 409-17 DOI 10.2450/2019.0245-19).
However, we think that in these particular patients with AIIRD, about whom in the revised version of the manuscript the possible reasons of the quite high prevalence of HBV markers have widely been discussed (lines 206-223), the HBV prevalence has not substantially changed for a series of considerations. The effects of HBV vaccination are still poorly influent, considering that the large majority of the described patients are out of the age target of the compulsory vaccination and only in the next 2-3 decades a population of this mean age will be reached by the compulsory vaccination. Moreover, with the availability of biologics and more recently of the kinase inhibitors, and the maintaining of corticosteroids, the rate of immunosuppression induced by the treatment of the AIIRD patients has not been reduced. Finally, the sensitivity of the rheumatologists, compared to hepatologists, regarding the need for compulsory screening for HBV in patients undergoing immunosuppressive therapy and the need to vaccinate the seronegative patients, in order to protect this vulnerable population, is lower. Thus, although the study was conducted in a small area of Italy and in a relatively small population and recently an analytical review could not demonstrate any significant different prevalence of HBV infection between patients with AIIRD and general population (RMD Open 2019;5: e001041. doi:10.1136/rmdopen-2019-001041), nonetheless we think that the observed results may quite faithfully express the real prevalence of HBV infection in AIIRD patients.
.
Point 2: To improve the importance and informativeness of the study, I would suggest the Authors to update the results, if possible and available, with patients' follow-up data (eg if they were prescribed immunosuppressive or biological therapy and the outcomes in those with positive HBV markers). Additionally, they could add demographic (eg ethnicity) and clinical data (eg disease activity or disease duration) for completeness in Table 1.
Response 2: All patients have received immunosuppressive or biological therapy, as planned. Follow-up data have been provided in lines 266-270. Demographic and clinical data have been added to Table 1 and in the text (lines 142-144 and 190-193). Disease duration has been added, whereas disease activity has been calculated with an arbitrary and not validated index, considering that for not all the types of diseases a validated clinimetric index was available and that it is difficult to compare different clinimetric indices, so that it was not shown in the text. However, even with this not validated index, Chronic Arthritis patients had the significantly (p=0.007 vs Spondyloarthritis) highest disease activity index (2.33±0.68), whereas Connective Tissue Disease had 2.1±0.78 and Spondyloarthritis had 2.03±0.72. To substitute for disease activity index, inflammatory markers, such as CRP and ESR, have been added in the text and in the Table 3, in order to comply with the right indication of the reviewer.
Point 3: Finally, I do not understand the rationale of comparing proportions between the groups with statistical tests without an a priori hypothesis.
Response 3: The reviewer is right, the rationale of the comparison between the groups has not been clarified and we want to particularly thank the reviewer for having pushed us to better define the clinical aspects of the diagnostic groups, which has allowed to identify the Chronic Arthritis group as the patients with the highest prevalence of HBV and inflammation markers, two conditions which are probably not casually found together, as discussed in lines 237-240. Moreover, even the fact that Chronic Arthritis and Spondyloarthritis patients are those more frequently planned for biologic therapy and more frequently showing HBV infection markers, thus at higher risk of reactivation (lines 243-247), is an additional element to add to the rationale of looking for the prevalence of HBV markers in diagnostic groups.
Authors' Reply to the Review Report (Reviewer 2)
Point 1: Although well written and presented, it must be pointed out that the study has been conducted in 2012-2013 and the results may be quite outdated. As the Authors correctly state, universal HBV vaccination is nowadays very common in Italy and it has to be taken into account when presenting such dated results. Moreover, the study has been conducted in a small area of Italy and the prevalence estimates may not be very informative for practitioners in other countries.
Response 1: We thank the reviewer for his kind appreciation. Actually, the study may appear quite outdated having been realized in 2012-2013 and in Italy the prevalence of HBV markers in first time donors was
275.9×105 in 2009, declining to 143.6×105 in 2018 (Blood Transfus 2019; 17: 409-17 DOI 10.2450/2019.0245-19).
However, we think that in these particular patients with AIIRD, about whom in the revised version of the manuscript the possible reasons of the quite high prevalence of HBV markers have widely been discussed (lines 206-223), the HBV prevalence has not substantially changed for a series of considerations. The effects of HBV vaccination are still poorly influent, considering that the large majority of the described patients are out of the age target of the compulsory vaccination and only in the next 2-3 decades a population of this mean age will be reached by the compulsory vaccination. Moreover, with the availability of biologics and more recently of the kinase inhibitors, and the maintaining of corticosteroids, the rate of immunosuppression induced by the treatment of the AIIRD patients has not been reduced. Finally, the sensitivity of the rheumatologists, compared to hepatologists, regarding the need for compulsory screening for HBV in patients undergoing immunosuppressive therapy and the need to vaccinate the seronegative patients, in order to protect this vulnerable population, is lower. Thus, although the study was conducted in a small area of Italy and in a relatively small population and recently an analytical review could not demonstrate any significant different prevalence of HBV infection between patients with AIIRD and general population (RMD Open 2019;5: e001041. doi:10.1136/rmdopen-2019-001041), nonetheless we think that the observed results may quite faithfully express the real prevalence of HBV infection in AIIRD patients.
.
Point 2: To improve the importance and informativeness of the study, I would suggest the Authors to update the results, if possible and available, with patients' follow-up data (eg if they were prescribed immunosuppressive or biological therapy and the outcomes in those with positive HBV markers). Additionally, they could add demographic (eg ethnicity) and clinical data (eg disease activity or disease duration) for completeness in Table 1.
Response 2: All patients have received immunosuppressive or biological therapy, as planned. Follow-up data have been provided in lines 266-270. Demographic and clinical data have been added to Table 1 and in the text (lines 142-144 and 190-193). Disease duration has been added, whereas disease activity has been calculated with an arbitrary and not validated index, considering that for not all the types of diseases a validated clinimetric index was available and that it is difficult to compare different clinimetric indices, so that it was not shown in the text. However, even with this not validated index, Chronic Arthritis patients had the significantly (p=0.007 vs Spondyloarthritis) highest disease activity index (2.33±0.68), whereas Connective Tissue Disease had 2.1±0.78 and Spondyloarthritis had 2.03±0.72. To substitute for disease activity index, inflammatory markers, such as CRP and ESR, have been added in the text and in the Table 3, in order to comply with the right indication of the reviewer.
Point 3: Finally, I do not understand the rationale of comparing proportions between the groups with statistical tests without an a priori hypothesis.
Response 3: The reviewer is right, the rationale of the comparison between the groups has not been clarified and we want to particularly thank the reviewer for having pushed us to better define the clinical aspects of the diagnostic groups, which has allowed to identify the Chronic Arthritis group as the patients with the highest prevalence of HBV and inflammation markers, two conditions which are probably not casually found together, as discussed in lines 237-240. Moreover, even the fact that Chronic Arthritis and Spondyloarthritis patients are those more frequently planned for biologic therapy and more frequently showing HBV infection markers, thus at higher risk of reactivation (lines 243-247), is an additional element to add to the rationale of looking for the prevalence of HBV markers in diagnostic groups.

Reviewer 3 Report
The manuscript titled “Prevalence of Hepatitis B Virus Markers in Patients with Autoimmune Inflammatory Rheumatic Diseases in Italy” by Marco Canzoni et al., is an interesting survey describing the prevalence of HBV serum markers in patients affected by Autoimmune Inflammatory Rheumatic Diseases (AIIRD) in Italy. Considering the high prevalence of HBV infection in Italy and the risk of reactivation of HBV in patients receiving immune suppressive treatments, the paper gives new insights on this topic. More than 300 patients were included in the analysis and, after excluding vaccinated subjects, previous exposure to HBV occurred in 24% of AIIRD patients. Interestingly only 30% of them were aware of HBV previous infection.
Although the manuscript helps in clarifying the risk of HBV reactivation in AIIRD patients and summarizes current indications for the management of these patients, major revisions are required.
Line 105: “corticosteroids at the dose ≥7.5 mg/day”. Please clarify the type of corticosteroid given at 7.5 mg/day
Line 121: “Abadacept” should be written without capital letter, being the name of the molecule.
Line 124: Be consistent with S. Andrea University Hospital denomination throughout the text.
Line 135: “The sex ratio (M/F) was 0.6” please, rephrase (M to F sex ratio was 0.6)
Line 149: “he was counted twofold” please rephrase (he was counted twice)
Line 154: is the HBsAb titer measured in IU/ml? Or is it measured in IU/l or mUI/ml?
Line 155: “p<0,04” Express the significance level as p<0,05 or providing the exact value
Line 162: “None subject resulted” please rephrase
Lines 155-166: Please provide HBcAb/ABsAb serostatus of the four patients with detectable HBV-DNA (<200 IU/ml)
Line 167-171: “The prevalence of any HBV marker peaked in the group of Chronic Arthritis as compared to the other diagnostic groups, significantly (p<0.02) versus Connective Tissue Disease group, and was more likely observed in subjects older than 60 years as compared to those younger than this age (29% vs 19%; p<0.05). No statistical difference was observed by sex and by potential suitability for immunosuppressive therapy (Table 3)”. The sentence should be rephrased
Line 175: “p<0,04” Express the significance level as p<0,05 or providing the exact value
Line 196-198: “the two patient groups who are also those more frequently planned for biologic therapy (in fact, 45/48 patients planned for biologic therapy belonged to these diagnostic groups), thus at higher risk of reactivation”. The sentence should be rephrased.
Line 211-212: I wonder if the term prophylaxis is appropriate in the case of treatment of a patient with active HBV replication as demonstrated by HBV-DNA detection. Would it be better to use the term “therapy” instead?
Line 220 and 224: Please be consistent using HB vaccination instead of HBV vaccination
Line 242-246: “In fact, when the observed immune response to vaccine was primary, a false anti-HBc positivity was hypothesized, thus needing the complete vaccine cycle, whereas if the observed immune response was anamnestic, the anti-HBc positivity was attributed to a resolved infection, thus needing only one booster; finally, if the immune response was weak or lacking, anti-HBc positivity was attributed to an occult HBV infection”.
The relationship between the response to HBV vaccination and anti-HBcAb positivity is not completely explained in this sentence. I would suggest to rephrase in order to make the concept clearer. Citations may help.
Author Response
We wish to thank the reviewers for the constructive remarks which have allowed to substantially improve the manuscript and we apologize for 2 inaccuracies which have been noticed when revising the file, the first represented by the percentage of patients aware of their serological HBV marker situation, which has been reduced from 21/70 (30%) to 11/70 (16%), as a consequence of a more accurate check (line 255), and the second by the changes of 2 numbers in Table 2, which refer to different categories of HBV marker positivity, which do not modify the total amount of HBV positivity.
Authors' Reply to the Review Report (Reviewer 3)
Point 1: Line 105: “corticosteroids at the dose ≥7.5 mg/day”. Please clarify the type of corticosteroid given at 7.5 mg/day
Response 1: “corticosteroids at the dose ≥7.5 mg/day” has been replaced by “≥7.5 mg/day of prednisone or equivalent”
Point 2: Line 123: “Abadacept” should be written without capital letter, being the name of the molecule.
Response 2: “Abatacept” has been replaced by “abatacept”
Point 3: Line 126: Be consistent with S. Andrea University Hospital denomination throughout the text.
Response 3: We complied with the reviewer’s suggestion
Point 4: Line 140: “The sex ratio (M/F) was 0.6” please, rephrase (M to F sex ratio was 0.6)
Response 4: “The sex ratio (M/F) was 0.6” has been rephrased according to the reviewer’s indication
Point 5: Line 153: “he was counted twofold” please rephrase (he was counted twice)
Response 5: “he was counted twofold” has been replaced by “he was counted twice”; “twofold” has been replaced by “twice” also in Table 1
Point 6: Line 161: is the HBsAb titer measured in IU/ml? Or is it measured in IU/l or mUI/ml?
Response 6: We apologize for the mistake and we have modified “IU/ml” in “mIU/ml”
Point 7: Line 162: “p<0,04” Express the significance level as p<0,05 or providing the exact value
Response 7: “p<0.04” has been replaced by “p<0.03401”
Point 8: Line 178: “None subject resulted” please rephrase
Response 8: “None subject resulted” has been replaced by “No subject resulted”
Point 9: Please provide HBcAb/ABsAb serostatus of the four patients with detectable HBV-DNA (<200 IU/ml)
Response 9: “HBV-DNA was <200 IU/ml in four” has been replaced by “HBV-DNA was <200 IU/ml in four (2 isolated anti-HBc and 2 anti-HBc/anti-HBs positive subjects)” (lines 181-183).
Point 10: “The prevalence of any HBV marker peaked in the group of Chronic Arthritis as compared to the other diagnostic groups, significantly (p<0.02) versus Connective Tissue Disease group, and was more likely observed in subjects older than 60 years as compared to those younger than this age (29% vs 19%; p<0.05). No statistical difference was observed by sex and by potential suitability for immunosuppressive therapy (Table 3)”. The sentence should be rephrased
Response 10: “The prevalence of any HBV marker peaked in the group of Chronic Arthritis as compared to the other diagnostic groups, significantly (p<0.02) versus Connective Tissue Disease group, and was more likely observed in subjects older than 60 years as compared to those younger than this age (29% vs 19%; p<0.05). No statistical difference was observed by sex and by potential suitability for immunosuppressive therapy (Table 3)”, even taking into account the remarks of the reviewer N 2, has been replaced by “In the present survey HBV infection markers were more likely observed in patients with Chronic Arthritis, who were also those presenting the highest levels of inflammation markers, one condition which has been recently found to be predictive of a hyporesponse to HB vaccination [19], thus allowing to hypothesize that these patients are even less efficient to respond to and clear HBV. Moreover, HBV infection markers were more prevalent in subjects older than 60 years as compared to those younger than this age (29% vs 19%; p<0.05). No statistical difference was observed by sex and by potential suitability for immunosuppressive therapy, nor for disease duration among the different diagnostic groups.” (lines 236-243).
Point 11: “p<0,04” Express the significance level as p<0,05 or providing the exact value
Response 11: “p<0.04” has been replaced by “p=0.03455” even in footnote Table 4
Point 12: “the two patient groups who are also those more frequently planned for biologic therapy (in fact, 45/48 patients planned for biologic therapy belonged to these diagnostic groups), thus at higher risk of reactivation”. The sentence should be rephrased.
Response 12: The sentence has been rephrased as follows: “Finally, Chronic Arthritis and Spondyloarthritis patients, who had higher prevalence of HBV markers compared to Connective Tissue Disease patients, were more frequently planned for biologic therapy (in fact, out of 48 patients cumulatively planned for biologic therapy 45 [94%] belonged to these diagnostic groups), thus exposing them to a higher risk of reactivation.” (lines 243-247).
Point 13: I wonder if the term prophylaxis is appropriate in the case of treatment of a patient with active HBV replication as demonstrated by HBV-DNA detection. Would it be better to use the term “therapy” instead?
Response 13: Subjects with chronic HBV infection (those with HBV-DNA level <2000 IU/ml, i.e. former inactive carriers) receive prophylaxis; conversely therapy is administered to subjects with chronic HBV hepatitis according to the EASL 2017 document reported among the references.
Point 14:: Please be consistent using HB vaccination instead of HBV vaccination
Response 14: HB vaccination has been used in both, line 275 and 279.
Point 15: “In fact, when the observed immune response to vaccine was primary, a false anti-HBc positivity was hypothesized, thus needing the complete vaccine cycle, whereas if the observed immune response was anamnestic, the anti-HBc positivity was attributed to a resolved infection, thus needing only one booster; finally, if the immune response was weak or lacking, anti-HBc positivity was attributed to an occult HBV infection”.
The relationship between the response to HBV vaccination and anti-HBcAb positivity is not completely explained in this sentence. I would suggest to rephrase in order to make the concept clearer. Citations may help.
Response 15: The sentence has been rephrased, even by adding new citations, as follows: “In fact, when the observed immune response to vaccine was slow or primary, a relatively frequent condition in older studies [36-38,42], a false anti-HBc positivity was hypothesized, on the basis of the consideration that a true anti-HBc positivity witnesses a previous encounter with HBV able to prime the immune system, thus a rapid or anamnestic response would be expected; according to this interpretation, the anti-HBc positive subjects with slow response are interpreted as false positive and assimilated to seronegative subjects, thus needing the complete vaccine cycle. Conversely, if the observed immune response was rapid or anamnestic, the anti-HBc positivity was attributed to a resolved infection, with early loss of anti-HBs antibodies; according to this interpretation these patients only need one booster. Finally, if the immune response was weak or lacking, anti-HBc positivity was attributed to an occult HBV infection, in an individual unable to mount an effective immune response [45].” (lines 297-308).

Round 2
Reviewer 1 Report
The additional data provided and the deeper discussion of some issues make this interesting study worth of publication in its present form in Microorganisms.
Author Response
Thank you for your comment
Reviewer 2 Report
I appreciate the efforts of the Authors to amend the paper with the suggestions. However, I still find that there are several methodological issues. Additionally, the Discussion and Conclusion are not adequately supported by the Results.
Author Response
It is difficult for authors to understand the meaning of some sentences “several methodological issues” and “Additionally, the Discussion and Conclusion are not adequately supported by the Results”, as details are not specified and these remarks have not been raised in the previous revision.
We believed to having complied with the specific remarks of the first revision. Now we realize that our reply provided in the first revision was not reported even in the text and that the reviewer asked to modify only Table 1.
The Authors apologize for the misunderstanding and have provided to amend these points, generating a revised Table 1 and restoring the original Table 3. New paragraphs (lines 220-221, 224-232, 241-244 and 282-283) have been added regarding the queries of the reviewer, marked in red
Round 3
Reviewer 2 Report
I appreciate the efforts of the Authors to improve the manuscript.
By convention, significance levels should be reported only as "p<0.05, p<0.01, p<0.001"; if the Authors wish to retain the exact value, they should use p=... instead.
I have no further comments.